# A Gram Scale Soft-Template Synthesis of Heteroatom Doped Nanoporous Hollow Carbon Spheres for Oxygen Reduction Reaction

**DOI:** 10.3390/nano13182555

**Published:** 2023-09-13

**Authors:** Jisue Kang, Jong Gyeong Kim, Sunghoon Han, Youngin Cho, Chanho Pak

**Affiliations:** Graduate School of Energy Convergence, Institute of Integrated Technology, Gwangju Institute of Science and Technology, Gwangju 61005, Republic of Korea; jisue0419@gm.gist.ac.kr (J.K.); xaso123@gm.gist.ac.kr (J.G.K.); choxx576@gm.gist.ac.kr (Y.C.)

**Keywords:** nanoporous hollow carbon sphere, assisted soft template, nitrogen-doped, oxygen reduction reaction, cobalt-doped catalyst

## Abstract

Heteroatom-doped nanoporous carbon materials with unique hierarchical structures have been shown to be promising supports and catalysts for energy conversion; however, hard-template methods are limited by their inflexibility and time-consuming process. Soft-template methods have been suggested as an alternative, but they are limited by their picky requirements for stable reactions and the few known precursors for small-batch synthesis. In this study, a gram-scale soft-template-based silica-assisted method was investigated for producing nitrogen-doped hollow nanoporous carbon spheres (N-HNCS). Nitrogen doping is accomplished during preparation with enhanced electrocatalytic activity without complicating the methodology. To investigate the effect of the unique structural characteristics of N-HNCS (specific surface area: 1250 m^2^ g^−1^; pore volume: 1.2 cm^3^ g^−1^), cobalt was introduced as an active center for the oxygen reduction reaction. Finely tuned reaction conditions resulted in well-dispersed cobalt particles with minimal agglomeration. This sheds light on the advancement of new experimental procedures for developing more active and promising non-noble catalysts in large and stable batches.

## 1. Introduction

Nanoporous carbons are a type of carbon material that possess beneficial properties including high surface area, tunable pore sizes, large pore volumes, and diverse structures [1,2]. Due to their high thermal stability, good electric conductivity, and chemical inertness, nanoporous carbons are well-suited for a variety of applications in energy storage, conversion, and catalysis [3,4,5]. These physical and textural properties can be controlled by modifying their porous structures [6,7]. Heteroatoms such as nitrogen, as well as other atoms such as B, P, S, and O, are often incorporated into carbon structures to enhance their intrinsic physicochemical properties for energy and electrocatalytic applications, and this emphasizes the effect of carbon support [8,9,10,11,12,13]. Two common methods to fabricate the porosity of carbon materials are the hard-templating method and the soft-templating method (Figure 1a). Soft templates are constructed using amphiphilic surfactants and precursor moieties, while hard templates are created via nanocasting, where the structures of inorganic solid materials are replicated into porous carbon materials [14,15,16,17,18,19,20,21,22,23,24,25,26].

In recent years, significant efforts have been made to improve the soft-template method for fabricating hierarchically porous carbon materials, overcoming the limitations of the hard-templating approach. For example, the architecture of the hard-templating method is typically limited to the solid parent template, leading to uncontrollable outcomes during synthesis, so the use of tetraethylorthosilicate (TEOS) has been proposed as a valuable tool to assist in self-assembly while enabling the formation of mesopores in the final samples [27,28]. Several researchers have explored the application of diverse precursor materials to incorporate heteroatoms into the carbon framework using soft-templating and TEOS-assisted methods [29,30,31,32,33,34,35,36,37,38]. However, despite these advancements, existing studies on the TEOS-assisted method are predominantly conducted at the milligram scale, and there is a lack of comprehensive investigations into the effects of varying synthesis conditions on the resulting products. Therefore, there is a compelling need to conduct systematic investigations on synthesis conditions and scale up the production to the gram level.

Among the array of emerging carbon precursors, metal-organic frameworks (MOFs) have garnered attention for their exceptional ability to finely tune pore structures through their crystalline frameworks. Nevertheless, challenges such as the demanding conditions required for effective metal removal can impact both the properties and yields of the resulting carbon product, owing to the necessity of maintaining stability throughout carbonization. In light of these considerations, we turn our focus to the well-established resorcinol-formaldehyde (RF) resin as a viable carbon source. RF resins offer a more direct and accessible avenue, characterized by simplified template removal achieved via solvent extraction or thermal treatment. This avenue holds the potential to facilitate gram-scale production with commendable yields, thereby enhancing the practicality of our approach.

In this research, we aimed to optimize the TEOS-assisted synthesis of nitrogen-doped hollow nanoporous carbon spheres (N-HNCS) by precisely controlling the concentrations of melamine, resorcinol, and other variables to ensure reproducible and scalable synthesis at the gram scale. Additionally, we introduced cobalt metal ions as additives during the synthesis to create Co-doped N-HNCS. Due to the limited electrocatalytic ability of metal-free carbon nanomaterials, non-precious metal catalysts have been widely studied for the oxygen reduction reaction (ORR). Especially, Co-NC shows the structural stability of the ORR catalyst since it has the resistance of the Fenton reaction to produce high levels of radical oxygen species (ROS) to attack the carbon matrix, and it is difficult to separate Co atoms from the Co-N_x_ active site during the ORR process [39]. The focus was on designing hollow nanoporous carbon spheres and exploring possibilities as non-precious metal catalysts for ORR. We studied the catalytic performance of the resulting materials under both alkaline and acidic conditions. The inclusion of new additives and their effects on cobalt incorporation were thoroughly examined.

In conclusion, this research proposes a TEOS-assisted soft-template approach for producing N-HNCS, which contributes to the development of environmentally friendly and effective synthetic approaches to porous carbon materials for energy applications. The optimized synthesis method achieved reproducibility and scalability at the gram scale, addressing the limitations associated with hard-templating methods. The investigation of cobalt doping and the impact of various additives on the catalytic performance of N-HNCS demonstrate the potential of our synthesized materials as efficient catalysts for the ORR.

## 2. Materials and Methods

### 2.1. Synthesis of Nitrogen-Doped Hallow Nanoporous Carbon Sphere (N-HNCS)

In this investigation of N-HNCSs, a series of experiments were conducted using a silica-assisted soft template synthesis approach. The first step involved dissolving cetyltrimethylammonium bromide (CTAB, 5.558 g, 98%, Alfa Aesar, Ward Hill, MA, USA) in a solution of triple-deionized water (DIW, 180 mL) and ethanol (EtOH, 100 mL, DAEJUNG, Gyeonggi-do, Republic of Korea), followed by the addition of ammonia aqueous solution (NH_4_OH, 1 mL, extra pure, Junsei, Tokyo, Japan) and a premade solution of resorcinol (1.1 g, Alfa Aesar, 99%) and deionized water (20 mL). Then, tetraethylorthosilicate (TEOS, 6 mL, 95 wt.%, SAMCHUN, Seoul, Republic of Korea) and formaldehyde (1.48 mL, Alfa Aesar, 37% 2/2 aq. Soln., stab. with 7–8% methanol) were added to the reaction solution for complete dissolution, and melamine (0.6 g, Alfa Aesar, 99%) and formaldehyde (1.1 mL) were further added and stirred continuously for 24 h under the same conditions.

The resultant product, in the form of a dark orange solid, was filtered and washed with DIW and EtOH several times. The melamine-resorcinol-formaldehyde (MRF) resin-silica spheres obtained were air-dried overnight in a 60 °C oven and then ground into a fine powder using a mortar for further processing. Subsequently, the MRF resin-silica spheres were subjected to pyrolysis in an Ar atmosphere in a tube furnace. The pyrolysis process was carried out with a gradual increase in temperature at a rate of about 3 °C/min up to 350 °C, where it was held for 3 h. This phase serves to remove moisture, volatile compounds, and functional groups from the precursor material, establishing a clean starting point for carbonization. The temperature was then raised to 800 °C at a rate of 2.5 °C/min and maintained at this temperature for 2 h. Elevated-temperature pyrolysis facilitates the breakdown of organic molecules and the formation of a carbonaceous structure with well-defined porosity. Finally, the black product was cooled to room temperature.

In order to eliminate silica from the obtained product, it was dispersed in a solution containing HF (49%, Fisher Chemical, Waltham, MA, USA), deionized water, and EtOH, with a specific volume ratio of 1:4.5:4.5. The mixture was stirred for 1.5 h, followed by filtration and multiple washing steps using deionized water and ethanol. The end product was then dried in an oven maintained at 60 °C and named N-HNCS, with a weight of 1.1 g (Appendix A).

In order to determine the optimal amount of resorcinol to use in the reaction, experiments were conducted using 1 mL, 5 mL, and 20 mL of the reagent. Based on the results, the amount of total DIW was adjusted to 200 mL for all reactions. The resulting samples were labeled as N-HNCS-1 mL, N-HNCS-5 mL, and N-HNCS-20 mL, or by 1 mL, 5 mL, and 20 mL in some cases. An additional sample was made without the dissolving process, and it was called N-HNCS-solid. In addition, variable test samples were prepared using different ratios of DIW and EtOH when CTAB was dissolved. The ratios of DIW to EtOH used for comparison were 200 mL:80 mL and 200 mL:60 mL, in addition to the main 200 mL:100 mL samples. Each carbon sample was identified by its DIW:EtOH ratio for comparison purposes.

### 2.2. Synthesis of Cobalt and Nitrogen-Doped Hallow Nanoporous Carbon Sphere (Co-N-HNCS)

The process for synthesizing cobalt and nitrogen-doped hollow nanoporous carbon spheres was based on the N-HNCS procedure. Initially, a solution consisting of NH_4_OH (1 mL), DIW (180 mL), EtOH (80 mL), and CTAB (5.558 g) was heated at 70 °C and stirred. Then, a premade solution of resorcinol (1.1 g) and DI (20 mL) was added and stirred for 30 min. Cobalt was introduced via an 8 wt.% solution of cobalt chloride hexahydrate (CoCl_2_·6H_2_O, 4 g, Alfa Aesar, 98%) in EtOH (46 g) that was mixed at a mole ratio of 1:2:2 with 8-hydroxyquinoline (8-HQ, 0.15 g, Alfa Aesar, 99%) and 1,10-phenanthroline monohydrate (1,10P, 0.186 g, 99%, Guaranteed Reagent, DAEJUNG, Siheung, Republic of Korea) until fully dissolved. This solution was then added to the reaction mixture and stirred for 30 min. Subsequently, TEOS (6 mL) and formalin (1.48 mL) were added, followed by melamine (0.6 g) and more formalin (1.1 mL), and the mixture was stirred for 24 h. The reaction mixture was filtered, washed, and dried, then ground into powder and pyrolyzed under the same conditions as N-HNCS. The resulting product was subjected to HF etching, washing, and drying, following the same steps as N-HNCS. Finally, the product was named Co-N-HNCS.

In order to investigate the effects of 8-HQ and 1,10P on the synthesized carbon samples, separate samples were produced with only one of these reagents and with neither reagent. These samples were identified as Co-N-HNCS-x, where x represents the reagent used (x = 8HQ, 1,10P, and “none”).

### 2.3. Analysis of Physicochemical Properties

The surface properties of the synthesized carbon materials were investigated using various techniques. Field emission scanning electron microscopy (FE-SEM, S4700, Hitachi, Tokyo, Japan) and ultra-high resolution field emission scanning electron microscopy (UH-FE-SEM, Verios 5 UC; Thermo Fisher Scientific, Waltham, MA, USA) were utilized to examine the surface characteristics and pore architectures. The transmission electron microscopy (TEM; Tecnai G2 F30 S-Twin; FEI Company, Hillsboro, OR, USA) technique was used to observe the material morphology, and energy-dispersive spectroscopy (EDS) mappings were obtained using the Oxford Ultim Max 65 instrument. The physical properties of the N-HNCSs were analyzed using nitrogen (N_2_) adsorption isotherms, which were measured using a Belsorp MAX (Microtrac MRB, Osaka, Japan) instrument at a low temperature with liquid nitrogen (77 K). The Brunauer-Emmett-Teller (BET) method was applied to calculate the specific surface area, and the pore size distribution (PSD) was determined via non-local density functional theory (NLDFT). The N-HNCSs structure was examined by X-ray diffractometry (XRD, SmartLab; Rigaku, Tokyo, Japan), and the electron states of components on the surface of the N-HNCSs were examined using X-ray photoelectron spectroscopy (XPS, NEXSA; Thermo Fisher Scientific, Waltham, MA, USA). Finally, the proportions of elements in the N-HNCSs were determined using an element analyzer (UNICUBE; Elementar, Hesse, Germany).

## 3. Results

### 3.1. Synthesis of Nitrogen-Doped Hollow Nanoporous Carbon Spheres (N-HNCS)

We present soft-templating methods and variables that control the morphology and stability of the overall reaction, using a well-known resorcinol-formaldehyde resin as the carbon source, CTAB as the surfactant, and TEOS as the silica precursor (Figure 1b). Emulsion droplets are formed via hydrogen bonding with the solvents, followed by electrostatic interactions between surfactants and silicates on the surface of the droplets. Carbon precursors are then polymerized to secure the structure before final carbonization.

#### 3.1.1. Effect of Resorcinol Solution Concentration

The success of the liquid-phase reaction in producing the desired product is influenced by the stability of the solution, particularly when new reagents are introduced, such as solid-state chemicals. In this study, melamine and resorcinol were used as solid-phase reagents, but the low solubility of melamine (3240 mg/L in DIW) in the base solvents used in the reaction made it unsuitable. On the other hand, resorcinol has a high solubility in water (1000 mg/mL in DIW), but its solid flake form may lead to inconsistencies in the final product. Therefore, experiments were conducted to investigate the effect of stabilizing resorcinol as an aqueous solution on the production of carbon composites. Different amounts of resorcinol were dissolved in varying volumes of deionized water while keeping the total reaction solvent constant at 200 mL.

SEM data indicate that decreasing the concentration of the resorcinol additive leads to a trend in synthesis, with all samples showing uniformity in size and shape within their own domains. TEM images reveal that as the concentration of the added solution increases, the overall outer structure becomes distorted, with the smooth surface of N-HNCS-20 mL becoming rougher in N-HNCS-5 mL and resembling a rambutan configuration in N-HNCS-1 mL (Figure 2a–c). Nitrogen adsorption-desorption isotherms display type IV curves, indicating mesoporous structures are closer to ink-bottle-type pore shapes with decreasing resorcinol solution concentration (Figure 2g,h). In addition, the curves are in accordance with the calculated BET surface areas (Appendix A). This shows the chunky-architecture, bingo chip-like N-HNCS-1 mL has more effect on the porous structure than the 5 mL sample, which showed 710 m^2^/g compared to 1300 m^2^/g. Pore size distributions show overall common peaks between 1 mL and 20 mL samples but with different intensities, with 2.8 nm pores being dominant but pores around 20 nm also being observed. Large pore size distributions in N-HNCS-1 mL are expected to come from the surface morphologies and from the frameworks between the cavity and the smooth surface in N-HNCS-20 mL, according to previous TEM analysis.

The XRD patterns exhibited a broad peak at 25°, indicating the presence of the (002) plane of amorphous carbon (Appendix A). Additionally, weak and broad peaks corresponding to the (100) and (110) planes were observed at 42° and 78°, respectively. It is worth noting that there were no significant differences observed in the peak patterns between the different N-HNCS samples.

XPS surveys and N 1s spectrum analysis were performed to investigate the bonding configuration and elemental composition, as shown in Appendix A. The N 1s XPS spectrum was fitted with four peaks: oxidized N (402.3 eV), quaternary N (400.7 eV), pyrrolic N (399.3 eV), and pyridinic N (398.3 eV), shown in Figure 2. These peaks correspond to different nitrogen configurations, indicating successful nitrogen doping in the samples (refer to Appendix A). Elemental analysis results did not reveal any clear trends between the concentrations and N/C ratios. The N-HNCS-20 mL sample had a nitrogen content of 2.0%, N-HNCS-5 mL had 2.6%, and N-HNCS-1 mL had 2.1% (see Appendix A). However, these results were found to be consistent with the XPS analysis, providing evidence for the impact of using melamine as a nitrogen dopant.

By analyzing N-HNCS-20 mL, N-HNCS-5 mL, and N-HNCS-10 mL, we can infer the effect of resorcinol solution concentration. Decreasing the resorcinol concentration or increasing the amount of DIW used for dissolution promotes a more homogeneous dispersion of resorcinol molecules compared to when they are in bulk solids or at higher concentrations. This condition facilitates the co-assembly of organic and inorganic phases with the assistance of silica, allowing resorcinol better access to other monomers for cross-linking into MRF resins. The improved dispersion of resorcinol contributes to the stabilization of particle synthesis. As a result, N-HNCS-20 mL exhibits a smoother exterior and a larger particle size. Conversely, as the concentration increases, rapid reactions occur around the added resorcinol before it can fully disperse in the reaction solution. This leads to rougher and chunkier surfaces in N-HNCS. This hypothesis is supported by SEM images, which demonstrate nearly identical surface morphologies between N-HNCS-1 mL and N-HNCS-solid (refer to Appendix A).

#### 3.1.2. Effect of Resorcinol Solvent Ratio

The impact of solvent ratios on the outcome of soft-templating synthesis is widely acknowledged [40]. In the silica-assisted soft-templating method employed in this study, the solvent consists of a total of 200 of mL DIW and 100 mL of EtOH. The selection of this particular ratio was based on a series of experiments conducted using varying amounts of ethanol.

To compare the effects of different resorcinol solution concentrations, the solvent ratio of DIW to EtOH was adjusted to 200:80 mL. SEM images revealed that higher resorcinol concentrations did not result in rougher surfaces, as observed in the 200:100 mL solvent ratio (Appendix A). Unexpectedly, N-HNCS-1 mL in the 200:80 mL solvent displayed smooth and relatively uniform exteriors, with particle diameters of approximately 350 nm. N-HNCS-5 mL in the 200:80 mL solvent exhibited a similar crumpled form as the 5 mL sample in the 200:100 mL solvent, but with an even distribution of particle sizes. The samples prepared with less ethanol still exhibited smoother surfaces compared to the 200:100 mL N-HNCS-5 mL samples mentioned earlier. As expected, the 20 mL samples in the 200:80 mL solvent showed smooth surfaces, but the particle size distribution was broader compared to the same 20 mL samples prepared in the 200:100 mL solvent. The particle sizes ranged from 100 to 300 nm, with the majority of N-HNCSs around 300 nm, occasionally including smaller particles. However, TEM analysis revealed notable differences in the inner structures of these 200:80 mL samples. While 20 mL N-HNCSs exhibited a hollow structure, 1 mL N-HNCSs appeared to be somewhat filled rather than hollow, with varying degrees of filling indicated by differences in brightness. Additionally, 1 mL N-HNCSs had thin shells, and the filling ranged from lightly filled to more densely filled particles, although the majority displayed a slightly filled morphology. The 5 mL samples resembled those prepared in the 200:100 mL solvent but with thinner shells. Based on the comparison among the three samples, the 20 mL sample was selected as the control group for further solvent ratio testing due to its successful formation of a hollow structure and overall particle morphology.

Subsequent experiments were conducted using a 20 mL resorcinol solution and varying solution ratios of 200:60, 200:80, and 200:100. SEM observations revealed a clear trend in particle stabilization as the amount of EtOH increased. In the case of the 200:100 mL ratio, the particles exhibited a uniform size of around 300 nm and maintained a hollow structure (Figure 3a). For the 200:80 mL ratio, the formation of spheres was less stable, as indicated by the uneven particle sizes (Figure 3b). Nevertheless, they still retained a hollow structure, albeit with larger and more distinct cavities. Insufficient EtOH resulted in the failure to produce N-HNCSs, as observed in TEM and SEM images, where small and irregular spheres, as well as structures that did not form spheres, were observed. These carbon spheres, when formed, had sizes of approximately 50 nm and lacked a hollow cavity (Figure 3c).

The N_2_ adsorption-desorption isotherm curves obtained exhibit type IV characteristics, indicating the presence of mesopores characterized by capillary condensation and resulting in hysteresis (Figure 3g). Notably, the 200:80 ratio shows a significant hysteresis phenomenon, suggesting the presence of ink bottle-shaped pores, possibly due to the large cavities within the shell structure. The pore size distribution curves exhibit main peaks at 2.8 nm for the 200:100 ratio and 2.9 nm for the other samples (Figure 3h). In terms of the larger mesopore region, both the 200:100 and 200:60 ratios display peaks corresponding to pores in the range of 20–30 nm. However, considering the particle size of the 200:60 ratio, these peaks are seen to originate from the spaces between particles. Analysis of pore volumes using the NLDFT method reveals total pore volumes of 1.3 cm^3^ g^−1^, 1.2 cm^3^ g^−1^, and 1.5 cm^3^ g^−1^ for the 200:100, 200:80, and 200:60 ratios, respectively (Appendix A). The calculated BET surface areas for these samples are 1250 m^2^ g^−1^, 1200 m^2^ g^−1^, and 1400 m^2^ g^−1^, respectively.

The XRD patterns revealed the presence of broad peaks corresponding to the (002) and (100) planes of amorphous carbon at 25° and 42°, respectively. Additionally, there were very weak and broad peaks observed for the (110) plane at 78°. However, no significant differences were observed in the overall patterns of the three samples, indicating similar structural characteristics (Appendix A).

The utilization of melamine as the nitrogen precursor in the synthesis process was deemed successful based on the XPS survey spectra (Appendix A). All three samples exhibited a nitrogen content of over 1.0 at%, which increased from 1.17 to 1.9 at% as the amount of EtOH decreased. The surface elemental composition and qualitative information regarding the chemical environments were analyzed via N 1s spectra (Figure 3d–f). Among the N-HNCS samples, the 200:100 ratio exhibited the highest percentage of pyridinic nitrogen at 19.2%, while the 200:60 ratio had only 10.8% (Appendix A). Pyridinic nitrogen was the only nitrogen species that demonstrated a discernible trend among the N-HNCS samples. Furthermore, elemental analysis was conducted to confirm nitrogen doping, revealing nitrogen weight percentages of approximately 1.0, 1.95, and 1.75 for the 200:100, 200:80, and 200:60 ratios, respectively (Appendix A).

Based on the obtained results, we could propose a mechanism for the formation of carbon particles. The sample with the lowest EtOH concentration exhibited rapid reactions that hindered the formation of hollow structures, resulting in the formation of clusters of round-like particles that were unable to separate from each other. As the proportion of ethanol increased, the surface tension of the solution decreased, leading to changes in the reaction rate and the size of emulsion droplets. In the case of the 200:80 samples, larger cavities were observed compared to the 200:100 samples, and this can be attributed to the emulsion droplets. As the surface tension decreases, the size of the emulsion droplets also decreases since the dispersion of surface tension energies becomes less necessary. Consequently, the 200:100 samples exhibited multicavity structures instead of the larger interconnected cavities observed in the 200:80 samples. Additionally, the polymerization reaction rate decreased with an increase in ethanol concentration, forming larger and smoother samples [41]. Therefore, the 200:80 samples can be seen as a transitional phase between the other two phases when comparing particle size.

Overall, the choice of solvent ratios in the synthesis of N-HNCS had a significant influence on the overall process and outcomes. The 200:100 solvent ratio demonstrated the most favorable conditions, leading to the formation of stable and well-defined sphere particles. The selection of appropriate reaction solvents is crucial as they directly impact the self-assembly process involving resins, silicas, and surfactants. Achieving a balance in solvent ratios is important for obtaining consistent and reproducible results. In this study, the 200:100 N-HNCS sample exhibited the highest level of reproducibility, further emphasizing the importance of solvent selection in achieving desired synthesis outcomes.

### 3.2. Design of Cobalt and Nitrogen-Doped Hollow Nanoporous Carbon Spheres (Co-N-HNCS)

Building upon the successful silica-assisted soft-templating process described earlier, this study aimed to utilize N-HNCS as catalyst supports for the ORR. The focus was on developing non-precious metal catalysts, with cobalt chosen as the catalyst material due to its stability in acidic electrolytes and to avoid the Fenton reaction. Cobalt was incorporated into the synthesis process as an 8 wt.% precursor solution, similar to the inclusion of resorcinol, in order to facilitate the desired catalytic properties of N-HNCS.

Several tests were conducted to optimize the synthesis of Co-N-HNCS, including the determination of the appropriate step for adding the cobalt precursor. After numerous attempts, it was decided to disperse the cobalt precursors in the reaction solvent before the addition of TEOS. This approach leveraged the condensation of silicas to prevent the agglomeration of the metal suspensions, resulting in well-dispersed cobalt particles on the carbon structures. This in situ cobalt doping process eliminated the need for any additional post-treatment steps.

The newly introduced process for cobalt doping in Co-N-HNCS was validated via TEM imaging and EDS analysis. Initially, cobalt particles were not visually apparent, but upon closer inspection, small lattices were observed in regions that were initially perceived as carbon walls (Appendix A). The magnified areas indicated the presence of cobalt. Additionally, EDS mapping confirmed the dispersion of cobalt particles throughout the entire carbon sphere rather than being concentrated in specific regions (Figure 4b). These results provide conclusive evidence that the designed process successfully achieved cobalt doping in Co-N-HNCS.

As previously mentioned, this synthesis is based on the formation of emulsion droplets through hydrogen bonding and the further occurrence of electrostatic interactions that lead to cross-linking and hydrolysis polymerization. In this case, the reaction of the resins is somewhat faster than the sol-gel process of TEOS at the beginning. Thus, the cross-linking of resins and CTAB occurred through electrostatic interactions with small amounts of TEOS, forming a type of core before the shell. As the condensation and growth go on, the MRF emulsion concentration decreases, slowing down the overall process. The hydrolysis polymerization of silicates, on the other hand, prevails. The co-assembly of silicate oligomers with MRF emulsion droplets interact with CTAB to begin forming shells at the surface of the previously formed core structures.

The carbonization process conducted at 800 °C plays a crucial role in the removal of CTAB templates and the transformation of resin polymers into a carbon network that yields silica. This process involves the dehydrogenation of silica and the further polymerization of resin polymers, leading to a contraction or shrinkage of the porous carbon-silica composites. Due to the composition of the cores, which primarily consist of resins and less silica, the shrinkage effect is more pronounced in the cores compared to the shells, which contain more rigid silica frameworks. As a result, differential shrinkage occurs between the cores and shells, leading to their separation and the formation of the distinctive Co-N-HNCS configuration observed in the material.

After the carbon-silica composites are formed, an HF etching process is conducted to remove the silica framework. In the case of a sample subjected to a short HF etching time, TEM and EDS analysis revealed the presence of residual silica, providing insights into the condensation of TEOS during the synthesis. This contradicted our first hypothesis that silica nanospheres are initially formed in the core, and resin polymerization occurs at the surface of these spheres, resulting in the formation of cavities after HF etching. The EDS analysis of partially etched silica further supports this mechanism (Figure 4c), as the distribution of Si indicates that the concentration of silica gradually increases from the middle of the core towards the outer side of the particle. However, in the fully etched Co-N-HNCS sample, no excess traces of silica were observed via EDS analysis (Figure 4b). This suggests that the HF etching process effectively removes the silica framework, leaving behind a carbon-based structure with embedded cobalt.

Indeed, the addition of certain reagents and their specific conditions can have a significant impact on the formation and structure of Co-N-HNCS. In this study, while keeping the amounts of TEOS and CTAB fixed, variations in the addition of other reagents or the order of their addition likely influenced the reaction rates and subsequent formation of the Co-N-HNCS structure. These variations can affect the distribution and dispersion of cobalt particles within the carbon spheres as well as the morphology and properties of the resulting material. By carefully controlling the addition of reagents and their conditions, researchers can achieve the desired structure and properties of Co-N-HNCS for specific applications.

#### Effect of Additives on Electrochemical Activity

The addition of 8-hydroxyquinoline (8HQ) and 1,10-phenanthroline (1,10P) as chelating agents and nitrogen doping sources in the synthesis of Co-N-HNCS introduces interesting structural variations [42,43]. TEM imaging and XRD analysis (Appendix A) confirm that there are no cobalt aggregations and that the carbon frameworks are amorphous.

When 8HQ is used alone or in combination with other agents, the resulting Co-N-HNCS samples exhibit distinct yolk-shell structures with particle sizes of approximately 450 nm. The clear architecture of the shrunken centers indicates quick MRF resin cross-linking, resulting in the formation of the core. The broken particles reveal webbed walls of the cavity with random flakes and spaces between them. On the other hand, when only 1,10P is used, a flower-like hollow structure is observed in the Co-N-HNCS samples. The cavity borderline is somewhat unclear, but the brighter contrast in the center suggests a synthesis that grows outward with frameworks in a radial manner rather than core formation. These variations in structure highlight the influence of different reagents on the formation process, leading to the development of diverse Co-N-HNCS architectures. (Appendix A).

The nitrogen adsorption-desorption isotherms of the samples exhibit distinct characteristics and values in terms of pore analysis, despite their similar appearances (Appendix A). All samples display type IV hysteresis, indicating the presence of mesopores. However, only Co-N-HNCS-1,10P exhibits pore curves with a slit shape, while the other samples show ink bottle-like pore shapes. This observation aligns with the SEM and TEM images, which also reveal a unique structure for Co-N-HNCS-1,10P. The pore size distribution graphs demonstrate a common peak at 3.1 nm, with intensities corresponding to the gas adsorption amounts observed in the isotherms. Micropores of 1.4 nm are also observed, with Co-N-HNCS-8HQ displaying the highest intensity, consistent with the adsorption-desorption curves. The calculated BET surface area and pore volumes exhibit the same trend: Co-N-HNCS-8HQ > Co-N-HNCS > Co-N-HNCS-1,10P > Co-N-HNCS-none.

The effect of nitrogen doping using 1,10-phenanthroline is examined via elemental analysis (Appendix A). The samples with the addition of 1,10-phenanthroline exhibit higher nitrogen content, confirming its role as a dopant. Significant differences of up to 1 wt.% are observed among the samples, providing clear evidence of its effectiveness as a nitrogen dopant.

The activity of the samples was evaluated for the ORR at 1600 rpm under both acidic and alkaline conditions. In both 0.1 M HClO_4_ (acidic) and 0.1 M KOH (alkaline) electrolytes, the samples exhibited similar trends in activity. The order of performance was found to be Co-N-HNCS > Co-N-HNCS-1,10P > Co-N-HNCS-8 HQ > Co-N-HNCS-none. This suggests that the presence of Co-N-HNCS, as well as the addition of 1,10-phenanthroline and 8-hydroxyquinoline, positively influenced the ORR activity in both acidic and alkaline environments.

In acidic conditions, the presence of 1,10-phenanthroline demonstrates a significant impact on the ORR process, as evidenced by Figure 5. This effect can be attributed to the additional nitrogen doping provided by 1,10-phenanthroline. Similarly, the inclusion of 8-hydroxyquinoline as an additive also proves to be effective in enhancing the catalytic activity of the ORR, as clearly observed when comparing the curves to those of the Co-N-HNCS-none sample. Among the tested samples, Co-N-HNCS exhibits the highest activity. When compared at a fixed current density of 2 mA/cm^2^, Co-N-HNCS demonstrates a potential of 0.675 V, closely followed by Co-N-HNCS-1,10P at 0.656 V. In contrast, Co-N-HNCS-8-HQ and the none sample exhibit lower potentials of 0.572 V and 0.532 V, respectively.

In 0.1 M KOH, the overall performance of the evaluated samples is superior to that in 0.1 M HClO_4_, as depicted in Figure 5. When the current density is fixed at 2 mA/cm^2^, Co-N-HNCS exhibits the highest activity with a potential of 0.797 V. Among the samples containing only one additive, Co-N-HNCS-8HQ, and Co-N-HNCS-1,10P demonstrate similar activities, with potentials of 0.779 V and 0.789 V, respectively. The main sample, Co-N-HNCS-none, displays a potential of 0.750 V. The reaction rate in the oxygen diffusion zone is primarily determined by the oxygen diffusion rate, and the current density at which the plateau region first appears represents the limiting current density. In this study, after a few initial irregular peaks, the Co-N-HNCS samples reached limiting current densities of 4.28, 3.90, 4.95, and 3.58 mA cm^−2^ at 1600 rpm. Co-N-HNCS was less active for ORR catalysis in both alkaline and acidic media compared to commercial Pt/C; however, the effects of the chelating agent and N source were figured out.

The observed deep and broad peaks in all of the Co-N-HNCS samples can be attributed to multiple intertwined factors. Despite the presence of hierarchical structures, the Co-N-HNCS samples have a limited number of macropores, which results in inadequate mesopores and micropores for optimal mass transport. However, a rotating disk electrode (RDE) is not sufficient to investigate the effect of mass transport resistance; additional measurements using GDE would be applicable for further understanding [44]. Additionally, these performance evaluations were conducted on a glassy carbon electrode with a relatively high surface loading mass of 600 μg cm^−2^. The thick catalyst coating may contribute to the appearance of the deep hump between 0.7 and 0.8 V in alkaline conditions [21]. It is important to note that this study does not investigate rotating ring-disk electrode (RRDE) tests, which could provide insights into the ambiguous pathway of oxygen reduction. Many studies were performed on the Co-N-C catalysts using the Koutecky-Levich equation [39,45,46,47]. It is expected that the Co-N-HNCS catalyst exhibits both 2-electron and 4-electron reactions, leading to the observed curve shapes.

The electrochemical evaluations and characterizations conducted in this study demonstrate that the combined effect of these two additives leads to enhanced catalytic activity and improved overall performance. The individual presence of either 8-hydroxyquinoline or 1,10-phenanthroline alone does not yield the same level of performance as when they are used together. These findings highlight the synergistic effect of the two additives in optimizing ORR performance.

## 4. Discussion

Based on the findings of this investigation, the proposed method can be considered a TEOS-assisted soft template approach for synthesizing hollow carbon spheres with hierarchical pore structures. The synthesis process involves the use of water, ethanol, silicates, resorcinol, and formaldehyde, which form emulsion droplets through hydrogen bonding. TEOS acts as a silica precursor, assisting in the self-assembly of the MRF resins and surfactants. The strong electrostatic interactions between cationic CTAB molecules and anionic silicates enable the well-assembled surfactant micelles within the nanosphere frameworks. Overall, the ammonia-catalyzed polymerization of MRF resins resulted in the formation of uniform carbon spheres.

To optimize the synthesis process, studies were conducted to stabilize the added solid reagents. The concentration of pre-dissolved resorcinol was found to be most stable and reproducible when using a 20 mL solution. Similarly, a solvent ratio of DIW:EtOH = 200:100 was determined to be optimal. These variables collectively contributed to the successful synthesis of N-HNCS, with a good yield of nearly 1 g per batch.

During the characterization of cobalt-doped N-HNCSs, the role of silica was investigated. It was observed that silica was present not only in the less-etched core but throughout the entire particle. This finding led to the deduction that the specific formulation used in the synthesis process influenced the reaction speed. The sol-gel process of TEOS was initially faster than the resin reaction, resulting in the formation of a core through electrostatic interactions between the resins, CTAB, and a small amount of TEOS. As the condensation reaction progressed, the concentration of the MRF emulsion decreased, causing a slowdown in the overall procedure. In contrast, the silicate hydrolysis polymerization occurred more extensively, leading to the formation of shells on top of the core structures. The subsequent carbonization process utilized the differences in the dominant materials, resulting in varying degrees of framework shrinkage and the formation of the hollow structure. Nevertheless, there are still areas for further improvement. This silica-assisted synthesis method encounters the challenge of HF etching, which is a common issue with hard-templating methods. Future research should focus on finding environmentally friendly and straightforward approaches for the fabrication of carbon materials with hierarchical pore structures.

The dispersion of cobalt throughout the particle was observed, and the addition of reagents such as 8-hydroxyquinoline and 1,10-phenanthroline was found to enhance the ORR catalytic performance of the N-HNCS materials. It offers opportunities for continued development beyond the scope of this work, which primarily focuses on their basic catalytic applications. More in-depth studies are warranted to investigate the optimal catalyst structure, including the utilization of additional electrochemical measurements such as ORR with RRDE to gain insights into the reaction pathways. The concept of active sites is a key aspect in the field of Co-N-C catalysts, and further analysis of coordination via techniques like X-ray absorption spectroscopy (XAS) is crucial for the construction of efficient ORR catalysts. While this work employed chelating agents such as 1,10-phenanthroline and 8-hydroxyquinoline to form and maintain CoN_x_ moieties, the exact coordination environment was not confirmed. Therefore, more studies are required to elucidate this coordination, potentially via XAS analysis. Additionally, the optimal amount of cobalt loading and the calculation of actual loading amounts need to be thoroughly examined. Only by gaining a comprehensive understanding of cobalt loading effects can further analogies be made regarding the impact of cobalt. Through these incremental steps, a more effective design of nanoporous carbon can be achieved, which can be applied to various ORR applications in emerging energy solutions.

## 5. Conclusions

In conclusion, the optimized synthesis of N-HNCS and Co-N-HNCS demonstrates a TEOS-assisted soft-template approach. By utilizing the self-assembly process of soft templates, the particle shapes could be easily tuned by adjusting the reaction conditions. However, soft templates are known for their low yields and specific requirements. This work aims to bridge this gap by introducing TEOS into the reaction system. Silica serves as a pore-forming assistant and facilitates electrostatic interactions, thereby improving reaction stability and compensating for the low yields. The goal of achieving a stable and facile method for synthesizing nanoporous carbon spheres was successfully accomplished, along with the incorporation of nitrogen and cobalt doping.

## Figures and Tables

**Figure 1 nanomaterials-13-02555-f001:**
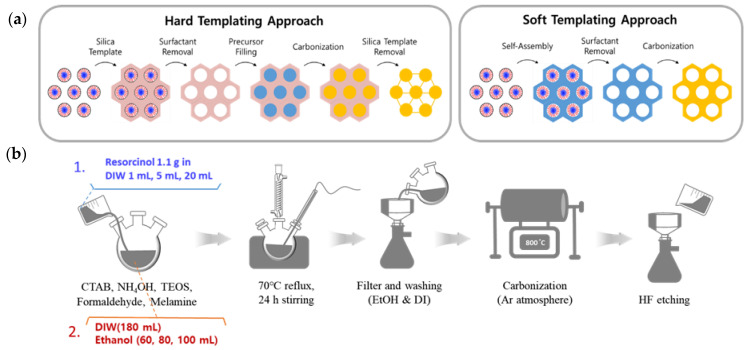
(**a**) A visualization of hard- and soft-templating methods and (**b**) Schematic procedures for synthesis of N-HNCS.

**Figure 2 nanomaterials-13-02555-f002:**
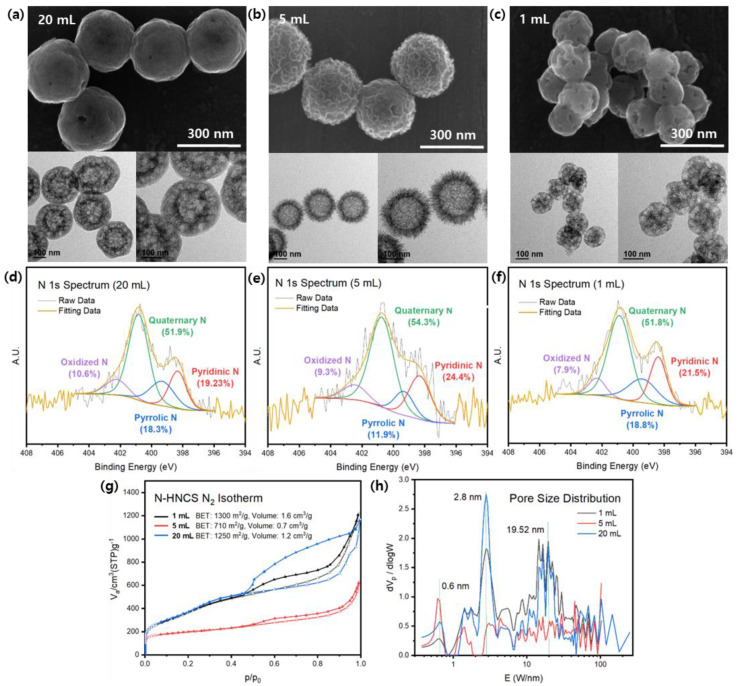
SEM, TEM images, and N 1s spectra of N-HNCS synthesized with different concentrations of resorcinol solution. (**a**,**d**) N-HNCS-20 mL, (**b**,**e**) N-HNCS-5 mL, (**c**,**f**) N-HNCS-1 mL. (**g**) Nitrogen adsorption isotherms of N-HNCS and (**h**) pore size distribution curves analyzed by NLDFT method.

**Figure 3 nanomaterials-13-02555-f003:**
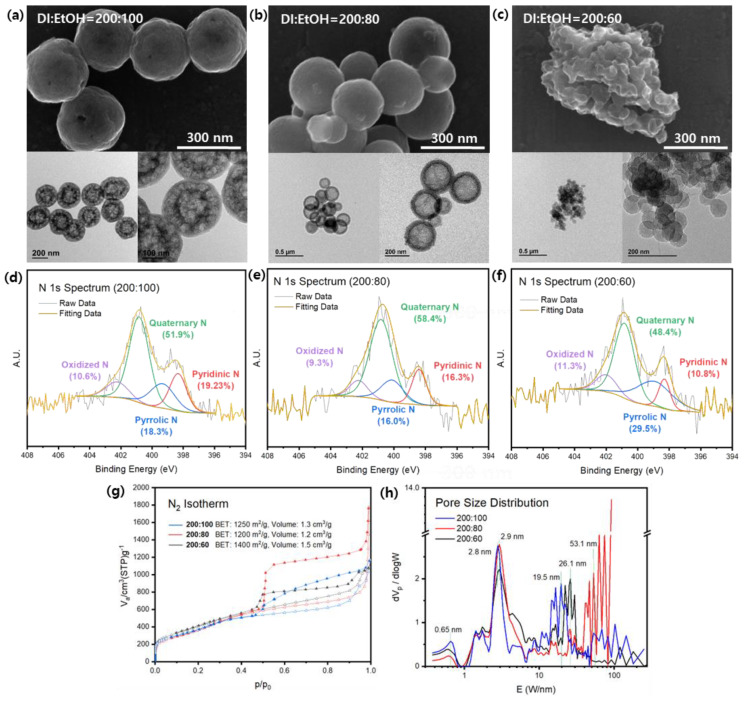
SEM, TEM images, and N 1s spectra of N-HNCS-20 mL at different solvent ratios. (**a**,**d**) DIW:EtOH = 200:100 mL, (**b**,**e**) DIW:EtOH = 200:80 mL, (**c**,**f**) DIW:EtOH = 200:60 mL, (**g**) nitrogen adsorption isotherms and (**h**) pore size distribution curves analyzed by NLDFT method of N-HNCS per solvent ratio.

**Figure 4 nanomaterials-13-02555-f004:**
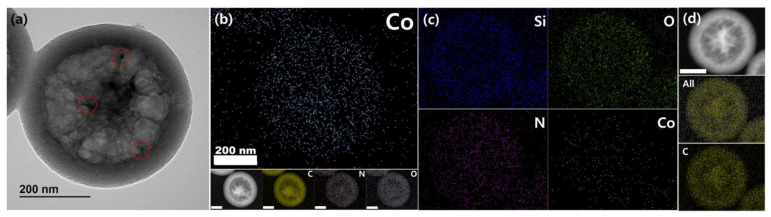
TEM analysis of the cobalt lattices of Co-N-HNCS. (**a**) An image of a single particle of Co-N-HNCS. Places where cobalt lattices were observed are circled in red. (**b**) Elemental mapping of Co, C, N, O and the original HAADF-STEM image; the scale bar represents 200 nm. EDS analysis of half-etched Co-N-HNCS. (**c**) Elemental mapping of Si, O, N, Co, C, and (**d**) the original HAADF-STEM image; the scale marker corresponds to 200 nm.

**Figure 5 nanomaterials-13-02555-f005:**
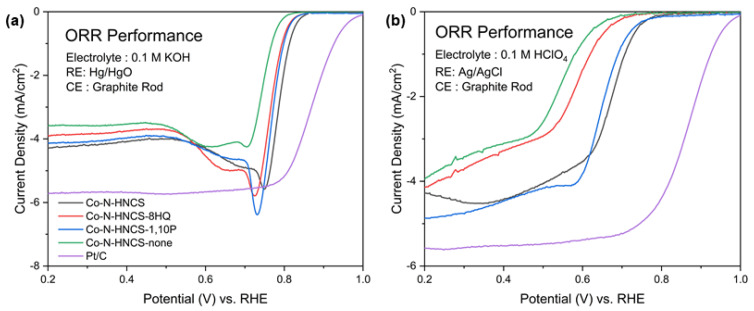
Half cell (three-electrode configuration) test (**a**) in acidic media, 0.1 M HClO_4_, and (**b**) in alkaline media, 0.1 M KOH electrolyte, saturated with O_2_ at 1600 rpm for the oxygen reduction reaction performance of Co-N-HNCS catalysts: Co-N-HNCS, Co-N-HNCS-8 HQ, Co-N-HNCS-1,10P, and Co-N-HNCS-none.

## Data Availability

The data presented in this study are available on request from the corresponding author.

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
