# Peer review of "A Gram Scale Soft-Template Synthesis of Heteroatom Doped Nanoporous Hollow Carbon Spheres for Oxygen Reduction Reaction"

_nanomaterials, 2023, doi:10.3390/nano13182555_

Round 1

Reviewer 1 Report

In this manuscript, the authors proposed a TEOS-assisted soft-template approach for producing hierarchical nanoporous carbon spheres. Authors claimed that they achieved reproducibility and scalability at the gram scale, addressing the limitations associated with hard templating methods. The work could be interesting for the scale-up synthesis of nanoporous carbon to be employed as a catalyst or catalyst support for various important reactions in renewable energy technologies. However, various concerns must be addressed before considering the manuscript for publication. The following are some comments for authors.

1.     Keywords should be revised with important and representative words and avoid repeating exact words, such as nanoporous carbon is repeated.

2.     I believe there are already reports on similar synthesis or even simpler synthetic routes to prepare porous carbon with silica as a template. How authors could convince the reviewer that this method is, novel and resultant nanoporous carbon is worthy to be prepared with this method. The silica-templated carbon precursors could be prepared in a very facile method in a beaker at room temperature with simple magnetic stirring why authors have selected the reflux method.

3.     In part 3.1.11, the authors discussed the Effect of Resorcinol Solution Concentration; it seems that of Resorcinol reagent makes the synthesis process complicated; why authors selected this reagent mainly, even though there are numerous carbon precursors available, such as MOFs-silica templates, which provide preexisting metal ions for doping as well. Were authors aware of those carbon precursors, as MOFs are the most emerging carbon precursors for heteroatom-doped carbon materials such as graphene or CNTs (reference Sci. Bull. 66, 21 (2021): 2207)? Authors should provide a discussion in the introduction.

4.     In the synthesis, The pyrolysis process was carried out with a gradual increase in temperature at a rate of about 3℃/min up to 350℃, where it was held for 3 h. The temperature was then raised to 800℃ at a rate of 2.5℃/min and maintained at this temperature for 2 h. Finally, the black product was  cooled to room temperature. Authors should explain why the annealing was performed in two steps and the impact on the final morphology of the carbon.

5.     Although authors claim a gram scale synthesis of the N-HNCS, no data support this claim. Authors should provide evidence to support their claims, possibly the digital images of the prepared N-HNCS during the weighing.

6.     Authors suggested to move the XPS results from SI to the main text; XPS is an important physical characterization for electronic structural analysis, especially for evidencing the doping or surface electronic properties of the synthesized nanoporous carbon.

7.     N-HNCS are important materials for ORR catalysis, yet the given ORR activities couldn’t be analyzed merely with LSV curves. Authors are suggested to provide detailed ORR activity and stability results for better evaluation of the prepared N-HNCS as catalyst support in electrocatalysis.

8.     There are some errors in the reference format. Authors are suggested to revise the references and update with some recent literature, such as 10.1016/j.trechm.2022.07.007, , 10.1039/D3QM00558E, 10.1002/anie.202115835

9.     Some spelling or grammatical errors should be revised.

Some spelling or grammatical errors should be revised.

Author Response

Letter to the Reviewers,

Before responding to the comments provided, we would like to express our sincere gratitude for the invaluable feedback provided by the reviewers during the initial review process, which has significantly contributed to enhancing the quality and comprehensiveness of our study.

In response to the thoughtful comments and suggestions made by the reviewers, we have meticulously revised the manuscript, diligently addressing each concern raised. We have seized the opportunity to update certain details in light of the latest developments in the field, ensuring the precision and relevance of our work. Answers to questions raised are answered in blue, and edits have been highlighted in yellow in the revised version of the paper.

With these revisions, we kindly request another opportunity for review and consideration, as we are enthusiastic about sharing our enhanced findings with the readership of your esteemed journal. Please rest assured that our manuscript is exclusive to Nanomaterials and has not been submitted or published elsewhere. We are confident that our research aligns seamlessly with the journal's scope and objectives.

Thank you once again for your invaluable guidance and consideration. We eagerly anticipate the chance to present our refined work to the Nanomaterials community.

Reviewer 2 Report

Overall, the quality of English is very good though some phrases are colloquial. Improvements in professional writing will help.

Author Response

(The authors gave the same response as above.)

Reviewer 3 Report

The article "A Gram Scale Soft-Template Synthesis of Heteroatom Doped Nanoporous Hollow Carbon Spheres for Oxygen Reduction Reaction" is devoted to obtaining highly efficient porous carbon carriers for use in the field of hydrogen energy. The article is devoted to a topical issue important for the development of materials for hydrogen energy. However, a number of remarks should be noted:

It is necessary to further investigate the obtained carbon materials by Raman spectroscopy.

When studying the activity of materials in ORR (Figure 6), it is necessary to add the results for a commercial Pt/C catalyst for comparison.

Why in Figure 6a the blue curve in the potential range of 0.9 - 1.0 V differs significantly from the other curves, what processes can occur in this potential range.

Why weren't measurements made at various rotational speeds to determine the number of electrons using the Koutecki-Levich equation?

Do you think it is possible to use the obtained carbon materials as carriers for platinum-containing catalysts?

Author Response

(The authors gave the same response as above.)

Round 2

Reviewer 1 Report

Manuscript seems to be improved. However, there are certain flaws to be addressed.

1. The ORR results in experimental details are not given. Authors are suggested to provide details about ORR measurements

2. The LSV curves in Fig 6 show author used Hg/HgO and Ag/AgCl reference electrodes in alkaline and acidic conditions. However, it is not clear whether the authors have converted the potential values to standard values such as reversible hydrogen electrode (RHE) or not. As mentioned Fig a, b, (Inset) the reference electrode is Hg/HgO and Ag/AgCl. Please clarify the legends and details about ORR measurements.

3. Authors are suggested to modify the conclusion. The conclusion should be a summary of the results. The future concerns about ongoing research should be discussed in the discussion section.

4. Fig 1 and Fig 2 could be merged into one figure for better presentation.

NA

Author Response

Please see the attachment including the responses to your comments.

Reviewer 2 Report

Please see the attached file for comments.

English language is adequate.

Author Response

Please see the attachment including the responses for your comments.

Reviewer 3 Report

The authors corrected most of the comments. However, on the issue «Why weren't measurements made at various rotational speeds to determine the number of electrons using the Koutecki-Levich equation?» We are not satisfied with the authors' response.

The point is that the article states «It is plausible that the Co-N-HNCS catalyst exhibits both 2-electron and 4-electron reactions, leading to the observed curve shapes»

Of course, such data can be obtained with a complex RRDE method, but the RDE using the Koutecki-Levich equation can also obtain these data, which does not require additional equipment. In addition, such measurements do not take much time and can be performed for several of the most interesting materials.

Author Response

Please see the attachment including the response to your comment.

Round 3

Reviewer 1 Report

N/A

NA